# Handgrip and gait metrics as scalable markers of physical health in schizophrenia and alcohol use disorders

Jaechan Park[1], Ye-Chan Kim[2], Kyujin Choi[2], Boram Park[2] and Hyun Gyu Lee[2] 

[1]Hyundai Sarang Hospital, Republic of Korea and [2]Inha University, Republic of Korea

## Research Article

schizophrenia; alcohol use disorder; relative handgrip strength; gait and functional mobility; wearable sensor assessment

**Corresponding authors:**
Hyun Gyu Lee and Boram Park;
Emails: hglee@inha.ac.kr; brpark@inha.ac.kr

J.P. and Y-C.K. contributed equally to this work.

## Abstract

Schizophrenia (SCZ) and alcohol use disorder (AUD) are associated with physical decline and motor dysfunction, but objective wearable-based motor assessments remain underutilized in psychiatric research. This study compared handgrip strength (HGS) and gait features between healthy controls (HCs) and individuals with SCZ or AUD using wearable sensors. A total of 434 participants (HCs: n = 210; AUD: n = 80; SCZ: n = 144) completed instrumented Timed Up and Go, walking, and HGS tests. Fifteen motor features were extracted and analyzed using multivariable linear regression adjusted for age, sex, and BMI. Five features—HGS, relative HGS (rHGS), walk quality index, symmetry index, and mid-turning phase duration—significantly differentiated one or both diagnostic groups from HCs. In AUD, rHGS showed moderate associations with multiple gait parameters, consistent with more widespread motor dysfunction. In SCZ, these associations were weaker, suggesting reduced coupling between upper- and lower-limb motor function. Both groups showed reduced HGS and gait alterations, but with distinct coordination patterns. These findings support wearable-based grip and gait metrics as scalable and objective motor functional markers in SCZ and AUD.

## Impact statement

Motor problems are common in severe mental illnesses, yet they are rarely measured in a systematic and objective way in routine psychiatric care. Most clinical assessments rely on symptom interviews, while subtle changes in physical performance often go unnoticed. This study demonstrates that simple, wearable-based measurements of grip strength and walking patterns can provide scalable and objective information about motor function in schizophrenia and alcohol use disorder. Importantly, our findings suggest that motor impairment is not uniform across psychiatric diagnoses. While both disorders show reduced physical performance compared with healthy individuals, the way upper- and lower-limb motor functions interact appears to differ between conditions. This distinction may have implications for rehabilitation strategies, physical health monitoring and risk assessment for long-term medical complications. Because grip and gait measures are inexpensive, noninvasive and easy to implement, they could be integrated into inpatient or outpatient settings as complementary tools to traditional psychiatric evaluation. Such approaches may support earlier detection of physical decline, guide personalized rehabilitation planning and contribute to more holistic care models that address both mental and physical health. Beyond schizophrenia and alcohol use disorder, this work highlights the broader potential of wearable technologies to quantify functional status in psychiatry. Objective motor metrics may become valuable markers for tracking recovery, treatment response and overall health in diverse psychiatric populations.

## Introduction

Schizophrenia (SCZ) and alcohol use disorder (AUD) are associated with a markedly reduced life expectancy – up to 20 years shorter than that of the general population – largely due to preventable physical health conditions, such as cardiovascular disease and falls, rather than psychiatric symptoms themselves (Ringen et al., 2018; Rogers et al., 2021). Individuals with these disorders face multifactorial risks: disease-specific factors (e.g., avolition, psychomotor slowing and extrapyramidal symptoms), environmental constraints (e.g., institutional confinement) and socioeconomic disadvantage contribute to their profound physical inactivity (Ringen et al., 2018; Rogers et al., 2021; Smail et al., 2023).

Recent research has increasingly recognized motor behavior as a central dimension in the pathophysiology of psychotic disorders. Work from leading groups, including Walther and Mittal, has conceptualized motor abnormalities as transdiagnostic markers reflecting disruptions across cortico–striato–cerebellar circuits (Moussa-Tooks et al., 2022; Walther and Mittal, 2022).

Objective and multimodal approaches – encompassing kinematic analysis, actigraphy and wearable sensor data – have demonstrated that psychosis-spectrum individuals exhibit persistent deficits in timing, coordination and sensorimotor integration, even outside acute episodes. Complementing these cross-sectional findings, longitudinal studies have linked psychomotor slowing to structural and functional alterations in basal ganglia and cerebellar networks and proposed its potential as a candidate endophenotype and biomarker (Osborne et al., 2020). Together, these findings provide a strong rationale for the quantitative evaluation of grip and gait performance in schizophrenia and related disorders.

SCZ and AUD differ in their underlying pathophysiology: SCZ is primarily neurodevelopmental, whereas AUD involves progressive neurodegenerative changes, particularly affecting cerebellar circuitry (Zhao et al., 2020). Despite these distinctions, both disorders share persistent motor abnormalities – such as deficits in gait, balance and muscle strength – that remain evident even during periods of clinical stability (Fein and Greenstein, 2013; Martin et al., 2022). These shared functional impairments suggest a potential common pathway of motor system dysfunction and underscore the need for objective, comparative evaluation of physical function in psychiatric populations.

Although declines in functional mobility (FM) have been documented in individuals with severe mental illness, objective assessments of FM remain underutilized in psychiatric care, particularly in low-resource or institutional settings (Firth et al., 2018; Morera-Salazar et al., 2021). Handgrip strength (HGS), gait analysis and the Timed Up and Go (TUG) test are validated, low-cost and scalable tools for assessing physical performance (Ogawa et al., 2022). Yet, most prior studies in psychiatric cohorts have relied on subjective evaluations or isolated metrics rather than integrated, sensor-based quantification.

Relative handgrip strength (rHGS) – normalized by BMI – has been suggested to be a more informative indicator than absolute HGS for predicting cardiometabolic risk, cognitive decline and all-cause mortality in large population studies (Li et al., 2018; Nygård et al., 2019; Chai et al., 2024). However, its clinical relevance in psychiatric populations remains underexplored. Studies examining HGS in AUD have yielded inconsistent findings: population-based studies report both detrimental and neutral effects of alcohol consumption on muscle strength, whereas clinical studies show persistent deficits (Kawamoto et al., 2018; Lee, 2021). In schizophrenia, motor dysfunction has also been widely reported, encompassing abnormalities in gait, balance and psychomotor speed, although findings vary across tasks and clinical states. Such discrepancies highlight the need for direct, instrumented comparisons between diagnostic groups using standardized wearable methods.

Moreover, psychiatric gait research has predominantly focused on aging, dementia or depression (Martin et al., 2022; Mengist et al., 2025). Few studies have directly compared SCZ and AUD with healthy controls (HCs) or examined the relationship between rHGS and detailed mobility parameters within these populations. Addressing these gaps, the present study simultaneously evaluated rHGS and quantitative gait features across three diagnostic groups (HC, AUD and SCZ) using wearable sensors to identify both disorder-specific and shared motor signatures.

Specifically, this study aimed to:

1. Compare absolute and relative HGS, gait parameters and iTUG sub-phase durations across groups; and

2. Examine within-group associations between rHGS and objective FM variables (gait parameters and iTUG sub-phase durations).

Wearable sensor technology enables fine-grained assessment of gait and iTUG performance in clinical and community settings (Martin et al., 2022; Ogawa et al., 2022). This approach aligns with current priorities for integrating physical health monitoring into psychiatric care, particularly in under-resourced settings where early detection and rehabilitation can improve long-term outcomes (Firth et al., 2018). Our findings aim to inform scalable, nonpharmacologic interventions for motor decline in SCZ and AUD and to advance data-driven, whole-person psychiatric care.

## Materials and methods

### Participants

The patient group (PT) comprised individuals diagnosed with alcohol use disorder (AUD) or schizophrenia (SCZ) according to the *Diagnostic and Statistical Manual of Mental Disorders, Fifth Edition, Text Revision* (DSM-5-TR; American Psychiatric Association, 2014). All participants were clinically stable and able to walk independently without assistive devices. Medication stabilization was defined as maintenance of the primary antipsychotic regimen for at least 2 weeks after the last dosage adjustment. This criterion reflects the final stabilization window, as all SCZ patients had been hospitalized for at least 8 weeks before assessment and demonstrated a median clinically stable period of 29.15 months.

Clinical stability was defined as follows:

- For both AUD and SCZ: Hospitalization for at least 8 weeks before assessment.
- For AUD: Absence of clinically relevant symptoms of alcohol intoxication or withdrawal during the assessment period.
- For SCZ: A stable clinical condition without acute psychotic exacerbation, maintained on a stable dose of the primary antipsychotic for at least 2 weeks.

A total of 252 inpatients and outpatients were initially recruited from closed and open psychiatric wards. Although education level was not formally recorded, literacy status was confirmed using medical records to ensure written consent capacity. One AUD patient who was illiterate but verbally consented was excluded due to comorbid rheumatoid arthritis affecting mobility. Thus, all participants in the final analysis were literate. Given South Korea's near-universal literacy rate among adults under 60 years, meaningful group differences in literacy were unlikely.

The healthy control (HC) group comprised 220 community and hospital staff participants, including nursing students from Jinju Health University and Moon Sung University, and staff members at the Uiryeong-gun Dementia Relief Center.

Exclusion criteria included:

1. Age under 19 years.
2. High fall risk during testing.
3. Comorbid diagnosis of both SCZ and AUD.
4. Acute exacerbation of psychiatric or physical illness.
5. History of conditions affecting gait or grip, such as severe hearing/vision impairment, Parkinson's disease, dementia, head trauma or major orthopedic surgery.
6. Presence of physical deformity, injury or acute pain.
7. Cognitive or intellectual impairment preventing comprehension of the study purpose, even after verbal explanation.

Of 472 initially enrolled participants (220 HCs and 252 PTs), 38 were excluded after detailed chart review – mostly due to previously unreported neurological or orthopedic conditions affecting mobility. The final analysis included 434 participants (Figure 1).

### Procedure and feature extraction protocol

#### Test procedures

Overview.  Participants first watched three demonstration videos (~3 min each) covering (1) flat-surface walking (12 m one way and 24 m round trip), (2) the Timed Up and Go (TUG) test and (3) grip strength measurement. One practice trial was provided for each task. All tests were conducted in the participant's familiar environment – hospital ward or home – to minimize performance variability.

Gait and TUG setup.  Gait and TUG tests were recorded using a BTS G-Sensor (BTS Bioengineering, Italy) attached to the sacrum (S1) for gait analysis and the second lumbar vertebra (L2) for TUG. The sensor incorporated a 3-axis accelerometer, gyroscope and magnetometer (sampling rate up to 1,000 Hz) and transmitted data via Bluetooth to G-Studio software on a laptop.

TUG procedure.  Participants began the TUG on a verbal "Go" cue while seated on a standard armchair (seat height 40 cm, width 40 cm, backrest height 38 cm, with armrests and a firm cushion). They stood up, walked 3 m, turned, returned and sat down. Five phases – sit-to-stand, forward walk, turning, return walk and stand-to-sit – were automatically segmented and analyzed (Janssen et al., 2002; Beauchet et al., 2011). A 1-min rest separated the gait and TUG assessments to allow sensor repositioning. The use of a standard chair with armrests followed the original TUG protocol (Podsiadlo and Richardson, 1991) and current CDC STEADI recommendations. Participants were instructed to walk at their natural, self-selected pace without external pacing.

**Fall risk management.** One trained researcher performed all sensor attachments and grip measurements to ensure consistency. A second assistant accompanied participants during walking tasks to prevent falls, especially among those with documented fall history or high fall risk (Baer and Ashburn, 1995). Enhanced supervision and safety precautions were applied in such cases.

Grip strength assessment.  Grip strength was measured using a digital dynamometer (T.K.K 5401, Takei Scientific Instruments, Japan). Participants stood upright with arms at their sides, and the handle width was adjusted to achieve a 90° angle at the second finger joint. Each hand was tested twice with 30-s rest intervals, and participants were instructed to squeeze maximally for 3 s. The highest of the four measurements was recorded. Relative handgrip strength (rHGS) was calculated as absolute grip strength divided by body mass index (BMI) (Muyor et al., 2022).

Feature extraction.  Fifteen quantitative features were extracted and grouped by domain:

- **Handgrip (2):** absolute HGS and relative HGS (rHGS);
- **Gait (7):** walking speed, stride length, walk ratio, walk quality index, symmetry index, propulsion index and propulsion index difference;
- **TUG (6):** total duration and durations of the five segmented phases. The full process is illustrated in Figure 2.

All feature definitions and measurement algorithms are detailed in Supplementary Table 1. A single valid trial per task was used for analysis, consistent with prior test–retest reliability studies (Yamada et al., 2019).

Ethical considerations.  The study was approved by the Institutional Review Board (IRB No. 240226-2A). All participants provided written informed consent. For participants unable to write, consent was obtained verbally after detailed explanation in a one-on-one session.

### Statistical analysis

Between-group comparisons were performed among healthy controls (HCs), alcohol use disorder (AUD) and schizophrenia (SCZ). Normally distributed continuous variables (age and body mass index [BMI]) were analyzed using one-way analysis of variance (ANOVA) and reported as mean ± standard deviation (SD). Nonnormally distributed variables (duration of illness, abstinent period in AUD, clinically stable period in SCZ and daily olanzapine-equivalent dose) were summarized as median (interquartile range [IQR]). Sex distribution was compared using the chi-squared test and expressed as count (percentage).

To examine group effects on motor function, 15 quantitative features (2 handgrip, 7 gait and 6 Timed Up and Go [TUG] variables) were analyzed using univariable and multivariable linear regression models, with HC as the reference group. The multivariable model was adjusted for age, sex and BMI. For analyses involving relative handgrip strength (rHGS), the multivariable model was adjusted for age and sex only, because rHGS was defined as handgrip strength normalized by body mass index (BMI). Regression results are

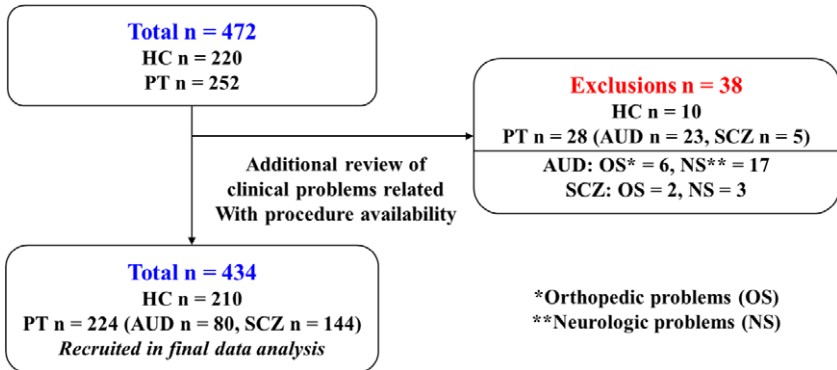

**Figure 1.** Flow diagram of participant selection.

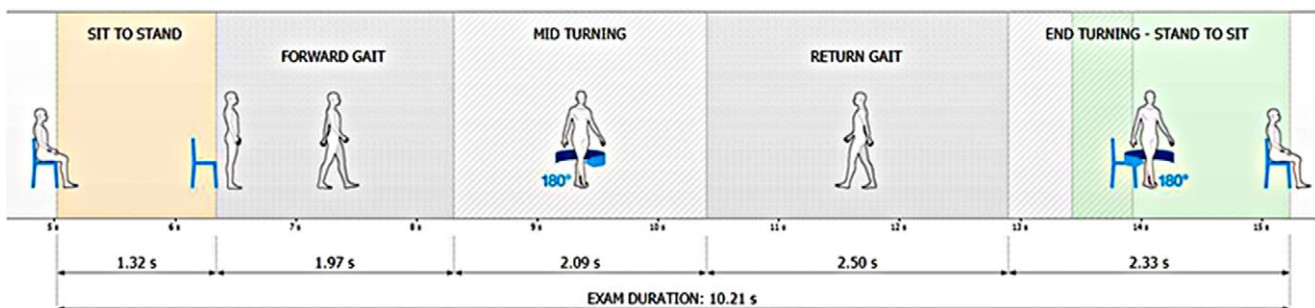

**Figure 2.** Test procedure. All values shown represent illustrative examples of segment-wise iTUG durations (seconds) obtained at a self-selected walking speed, without any externally imposed pacing.

presented as coefficients with standard errors (SEs) and 95% confidence intervals (CIs). To account for multiple comparisons across the 15 features, the false discovery rate (FDR) correction method was applied. Statistically significant variables from the uni- or multivariable models were visualized as mean ± SE bar graphs.

Associations between relative handgrip strength (rHGS) and six gait parameters (speed, stride length, walk ratio, walk quality index, symmetry index and total gait duration) were assessed using Pearson's correlation coefficients. Partial correlations were additionally computed, controlling for age, sex and BMI, and the results were visualized using scatter plots. To account for multiple comparisons across correlation analyses, false discovery rate (FDR) correction was applied, and adjusted *p*-values are reported.

Statistical significance was set at two-tailed $p < 0.05$. All analyses were conducted using SAS (version 9.3, SAS Institute Inc., Cary, NC, USA) and Python (version 3.9.18).

## Results

### Clinical characteristics

The final analysis included 210 healthy controls (HCs), 80 individuals with alcohol use disorder (AUD) and 144 with schizophrenia (SCZ). Group comparisons are summarized in Table 1.

Post-hoc comparisons indicated that both clinical groups were older than healthy controls, and BMI was higher in SCZ than in both HC and AUD. Sex distribution also differed, with a higher proportion of males in the AUD and SCZ groups compared with the HCs.

Among clinical variables, the median (IQR) duration of illness was 13.75 (8.35–24.92) years for AUD and 25.00 (16.00–33.00) years for SCZ. The median abstinent period in the AUD group was 17.18 (3.02–41.19) months, and the median clinically stable period in the SCZ group was 29.15 (7.89–65.75) months. The median daily olanzapine-equivalent dose in the SCZ group was 13.12 (7.87–24.82) mg/day.

### Linear regression model

Multivariable linear regression analyses identified five motor features showing significant group differences after false discovery rate (FDR) correction ($p < 0.05$, Table 2): handgrip strength (HGS), relative handgrip strength (rHGS), walk quality index, symmetry index and mid-turning phase duration. All multivariable models were adjusted for age, sex and body mass index (BMI), except for analyses involving rHGS, which were adjusted for age and sex only.

In these models, both diagnostic groups demonstrated reduced grip-related measures compared with healthy controls. Specifically, the estimated coefficients were as follows:

**Table 1.** Comparison of demographic and clinical characteristics

| Variable | | HC (*n* = 210) | AUD (*n* = 80) | SCZ (*n* = 144) | *p*-value |
|---|---|---|---|---|---|
| Age | Mean ± SD | 42.97 ± 16.25 | 58.42 ± 10.66 | 53.80 ± 11.00 | *<0.001* |
| BMI | Mean ± SD | 23.47 ± 3.95 | 23.74 ± 3.86 | 26.02 ± 4.35 | *<0.001* |
| Sex | Male | 80 (38.1%) | 178 (79.5%) | 104 (72.2%) | *<0.001* |
| | Female | 130 (61.9%) | 46 (20.5%) | 40 (27.8%) | |
| Total duration of illness (years) | Min–max | – | 0.44–49 | 1–47 | |
| | Median (IQR) | – | 13.75 (8.35, 24.92) | 25 (16, 33) | |
| Abstinent period in AUD (months) | Min–max | – | 1.87–115.36 | – | |
| | Median (IQR) | – | 17.18 (3.02, 41.19) | – | |
| Symptom stability period in SCZ (months)[a] | Min–max | – | – | 1.87–153.37 | |
| | median (IQR) | – | – | 29.15 (7.89, 65.75) | |
| Total olanzapine equivalents (mg/day) | Min–max | – | – | 0–92.15 | |
| | Median (IQR) | – | – | 13.12 (7.87, 24.82) | |

Abbreviations: AUD, alcohol use disorder; IQR, interquartile range (Q1, Q3); HC, healthy control; SCZ, schizophrenia; SD, standard deviation.
[a]Defined based on DSM-5-TR criteria as a clinically stable condition without acute symptom exacerbation.

- **AUD vs. HC:** lower HGS (−8.739), rHGS (−0.337) and symmetry index (−2.087).
- **SCZ vs. HC:** lower HGS (−9.917), rHGS (−0.383), walk quality index (−0.941) and symmetry index (−1.956), along with a prolonged mid-turning phase duration (+0.298).
- **SCZ vs. AUD:** higher rHGS was observed in SCZ (+0.116), with no other significant between-group differences.

Other gait- and Timed Up and Go–related features did not show significant group differences after covariate adjustment. The five significant features from the multivariable models are visualized in Figure 3 as mean ± standard error (SE) bar graphs. Univariable regression results are provided in the Supplementary Material for completeness.

### Correlation between rHGS and gait features

To examine associations between relative handgrip strength (rHGS) and gait parameters, partial correlations controlling for age, sex and BMI were calculated for six features – speed, stride length, walk ratio, walk quality index, symmetry index and analysis duration – and visualized using scatter plots. Statistical significance was determined after false discovery rate (FDR) correction to account for multiple comparisons. To assess the robustness of these findings, sensitivity analyses were conducted using Cook's distance to identify influential observations and an interquartile range (IQR)-based approach (1.5×IQR) to exclude extreme values. As shown in Supplementary Tables 5–7, exclusion of influential or extreme observations resulted in minimal changes in effect sizes and did not materially alter the statistical significance of the main findings, particularly for the AUD–HC comparisons.

In the AUD group, four parameters showed moderate correlations with rHGS: speed ($r = 0.3321$), stride length ($r = 0.4932$), walk ratio ($r = 0.4467$) and analysis duration ($r = -0.3917$). These findings indicate that greater grip strength was associated with faster and more efficient gait performance in AUD (Figure 4). In addition, sensitivity analyses using Cook's distance and IQR-based outlier exclusion demonstrated that the observed correlations between relative handgrip strength and gait features were robust to influential observations, with minimal changes in effect size or statistical significance (Supplementary Tables 5–7).

In contrast, the HC and SCZ groups showed only weak associations across all parameters ($r < 0.3$; Supplementary Figures 1 and 2), suggesting reduced coupling between upper- and lower-limb motor function in these groups.

### Discussion

This study quantitatively examined handgrip strength and gait using wearable sensors in schizophrenia (SCZ) and alcohol use disorder (AUD). Both groups showed marked reductions in grip strength and gait quality compared with healthy controls, yet they largely did not differ from each other in most motor features. Notably, coupling between rHGS and gait parameters was observed in AUD but was weak in SCZ, suggesting disorder-specific motor coordination patterns.

Together, these findings indicate that motor dysfunction represents a shared feature across SCZ and AUD, while the organization of motor coordination may differ between disorders. From a clinical perspective, this also suggests that sustained inpatient

stabilization and rehabilitation may help attenuate disorder-specific differences in motor function.

### Muscle strength deficits in SCZ and AUD

Both the AUD and SCZ groups demonstrated significantly lower absolute and relative handgrip strength (HGS and rHGS) than controls, with adjusted coefficients of −8.9 and −0.34 for AUD and −8.0 and −0.32 for SCZ. These deficits surpass the 1.6 kg threshold typically regarded as clinically meaningful and likely represent disorder-related neuromuscular pathology rather than demographic differences, as they persisted after adjustment for age, sex and BMI (Kim et al., 2018Bulbul et al., 2021; Martin et al., 2022).

In AUD, reduced muscle strength is consistent with the neurotoxic and myopathic consequences of chronic alcohol use (Fein and Greenstein, 2013; Zhao et al., 2020). Longitudinal evidence shows that heavy drinkers lose ~1 kg of HGS over 2 years (Cui et al., 2019), while mechanistic work implicates oxidative stress and mitochondrial dysfunction in alcohol-related myopathy (Simon et al., 2023). By contrast, population-based data remain inconsistent – moderate alcohol intake sometimes appears neutral or protective (Lee, 2021) – reflecting heterogeneity in drinking patterns, abstinence and metabolic health.

In SCZ, diminished HGS aligns with prior studies linking muscle strength to cognitive and functional capacity (Firth et al.2018). These results reinforce the view that muscle weakness is a core dimension of disease-related physical decline rather than a secondary medication effect.

### Gait impairments and subtle motor abnormalities

Both clinical groups exhibited reduced walk quality and gait symmetry, with coefficients of −0.76 and −3.28 in AUD and −0.94 and −2.12 in SCZ. Although symmetry indices remained within non-pathological ranges, these minor asymmetries are compatible with cerebellar involvement in AUD and coordination or extrapyramidal deficits in SCZ (Hirjak et al., 2018; Martin et al., 2022). Wearable sensors proved sensitive to these subclinical differences, identifying motor irregularities that might not be visible during bedside observation (Morera-Salazar et al., 2021).

### Functional mobility and task-specific deficits

Timed Up and Go (TUG) analysis revealed prolonged total duration in AUD (+0.72 s vs. HC) and extended forward-gait and mid-turning phases in SCZ (+0.18 s and +0.30 s). These distinct patterns point to cerebellar–vestibular dysfunction in AUD and psychomotor slowing in SCZ (Bombin et al., 2005; Zhao et al., 2020). Sit-to-stand performance was preserved, indicating that sequential motor integration – rather than general weakness – was primarily affected (Nuoffer et al., 2022).

Direct comparison showed no significant difference between SCZ and AUD in any feature (Table 2). The similarity may reflect shared mechanisms of motor recovery during long-term hospitalization, including sustained physical activity, pharmacologic stabilization and structured rehabilitation. Alternatively, subtle disorder-specific differences may persist below the detection threshold of current sensor-based measures. Integrating wearable metrics with neuroimaging or clinician-rated scales in future work could clarify the shared and distinct motor signatures of these disorders.

**Table 2.** Multivariable linear regression models

| | Outcome | Comparison | Multivariable model | | | |
| --- | --- | --- | --- | --- | --- | --- |
| | | | Coef. | 95% CI | $p$(raw) | $p$(FDR) |
| Handgrip strength | HGS (kg) | AUD-HC | −8.739 | (−10.919, −6.559) | <0.001 | <0.001 |
| | | SCZ-HC | −9.917 | (−11.686, −8.148) | <0.001 | <0.001 |
| | | SCZ-AUD | 1.178 | (−0.914, 3.269) | 0.269 | 0.457 |
| | rHGS (HGS/BMI) | AUD-HC | −0.338 | (−0.434, −0.242) | 0.000 | <0.001 |
| | | SCZ-HC | −0.454 | (−0.529, −0.378) | 0.000 | <0.001 |
| | | SCZ-AUD | 0.116 | (0.026, 0.206) | 0.012 | 0.049 |
| WALK | Speed (m/s) | AUD-HC | 0.002 | (−0.054, 0.059) | 0.931 | 0.931 |
| | | SCZ-HC | −0.032 | (−0.078, 0.014) | 0.173 | 0.338 |
| | | SCZ-AUD | 0.034 | (−0.020, 0.088) | 0.214 | 0.385 |
| | Stride length (m) | AUD-HC | −0.040 | (−0.087, 0.006) | 0.091 | 0.214 |
| | | SCZ-HC | −0.037 | (−0.075, 0.001) | 0.058 | 0.168 |
| | | SCZ-AUD | −0.004 | (−0.049, 0.041) | 0.875 | 0.916 |
| | Walk ratio (cm/steps/min) | AUD-HC | −0.027 | (−0.053, −0.002) | 0.036 | 0.125 |
| | | SCZ-HC | −0.012 | (−0.032, 0.009) | 0.274 | 0.457 |
| | | SCZ-AUD | −0.016 | (−0.040, 0.009) | 0.206 | 0.385 |
| | Walk quality index | AUD-HC | −0.764 | (−1.458, −0.069) | 0.031 | 0.125 |
| | | SCZ-HC | −0.941 | (−1.505, −0.378) | 0.001 | 0.008 |
| | | SCZ-AUD | 0.178 | (−0.488, 0.844) | 0.600 | 0.711 |
| | Symmetry index | AUD-HC | −2.087 | (−3.554, −0.620) | 0.005 | 0.030 |
| | | SCZ-HC | −1.956 | (−3.146, −0.766) | 0.001 | 0.009 |
| | | SCZ-AUD | −0.131 | (−1.538, 1.276) | 0.855 | 0.916 |
| | Propulsion index | AUD-HC | 0.169 | (−0.460, 0.799) | 0.597 | 0.711 |
| | | SCZ-HC | −0.090 | (−0.600, 0.421) | 0.730 | 0.801 |
| | | SCZ-AUD | 0.259 | (−0.345, 0.863) | 0.399 | 0.562 |
| | Propulsion index diff. | AUD-HC | 0.237 | (−0.038, 0.512) | 0.091 | 0.214 |
| | | SCZ-HC | 0.190 | (−0.033, 0.413) | 0.095 | 0.214 |
| | | SCZ-AUD | 0.047 | (−0.217, 0.311) | 0.727 | 0.801 |
| TUG | Analysis duration | AUD-HC | 0.394 | (−0.134, 0.921) | 0.144 | 0.294 |
| | | SCZ-HC | 0.514 | (0.085, 0.942) | 0.019 | 0.094 |
| | | SCZ-AUD | −0.120 | (−0.627, 0.386) | 0.641 | 0.740 |
| | Phases duration-sit to stand (s) | AUD-HC | −0.005 | (−0.092, 0.082) | 0.915 | 0.931 |
| | | SCZ-HC | 0.020 | (−0.050, 0.091) | 0.575 | 0.711 |
| | | SCZ-AUD | −0.025 | (−0.108, 0.059) | 0.558 | 0.711 |
| | Phases duration-forward gait (s) | AUD-HC | 0.097 | (−0.104, 0.299) | 0.342 | 0.513 |
| | | SCZ-HC | 0.181 | (0.018, 0.345) | 0.030 | 0.125 |
| | | SCZ-AUD | −0.084 | (−0.277, 0.109) | 0.394 | 0.562 |
| | Phases duration-mid turning (s) | AUD-HC | 0.153 | (−0.007, 0.312) | 0.060 | 0.168 |
| | | SCZ-HC | 0.298 | (0.168, 0.427) | <0.001 | <0.001 |
| | | SCZ-AUD | −0.145 | (−0.298, 0.008) | 0.063 | 0.168 |
| | Phases duration-return gate (s) | AUD-HC | 0.058 | (−0.159, 0.274) | 0.601 | 0.711 |
| | | SCZ-HC | −0.145 | (−0.321, 0.031) | 0.105 | 0.225 |
| | | SCZ-AUD | 0.203 | (−0.005, 0.411) | 0.056 | 0.168 |

*(Continued)*

**Table 2.** (*Continued*)

| Outcome | Comparison | Multivariable model | | | |
| --- | --- | --- | --- | --- | --- |
| | | Coef. | 95% CI | *p*(raw) | *p*(FDR) |
| Phases duration-stand to sit (s) | AUD-HC | 0.089 | (−0.095, 0.273) | 0.342 | 0.513 |
| | SCZ-HC | 0.160 | (0.011, 0.309) | 0.036 | 0.125 |
| | SCZ-AUD | −0.071 | (−0.247, 0.106) | 0.430 | 0.586 |

Note: The reference group in all regression models was the healthy control (HC) group. HGS: handgrip strength; rHGS: relative handgrip strength. All multivariable models were adjusted for age, sex and body mass index (BMI), except for rHGS models, which were adjusted for age and sex only.
Abbreviations: C, confidence interval; SE, standard error.

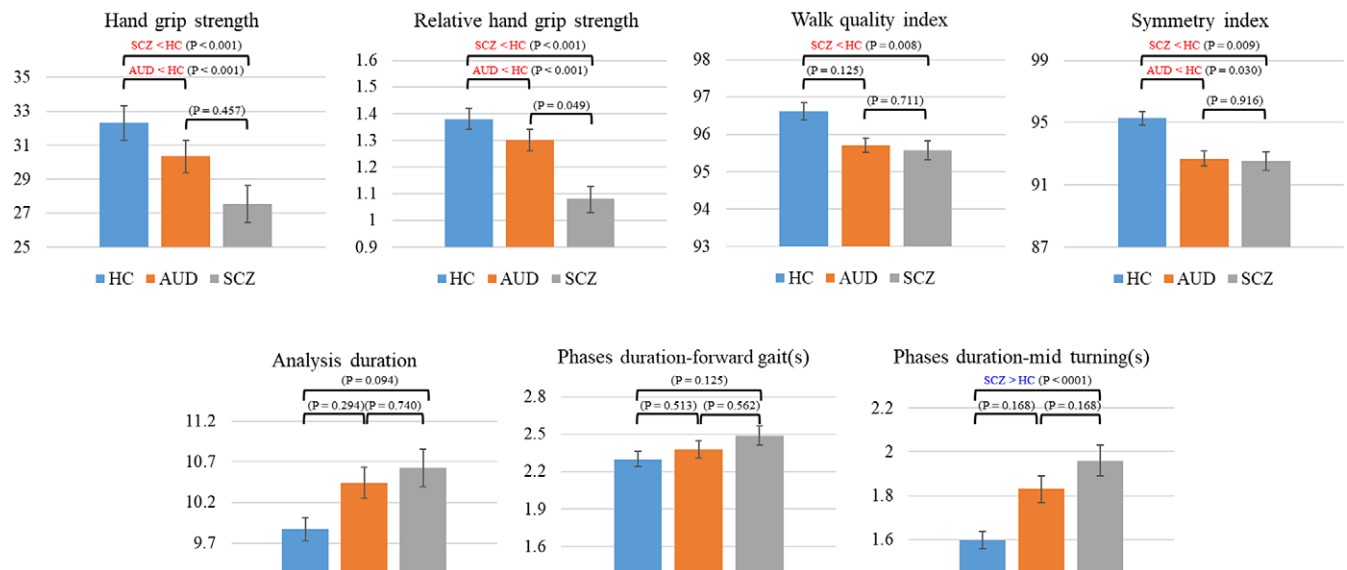

**Figure 3.** Mean and standard error for each feature after adjustment for age, sex and BMI across the three groups. Red and blue annotations indicate features that are significantly lower or higher, respectively, compared with the healthy control group. *p*-values for pairwise group comparisons, including schizophrenia versus alcohol use disorder, are derived from multivariable models and are reported after false discovery rate (FDR) correction.

### Linking grip strength and gait dynamics

Partial correlations revealed moderate associations between rHGS and gait parameters – speed, stride length, walk ratio and analysis duration – in AUD (*r* = 0.33–0.49). This suggests that greater muscle strength corresponds to faster, more efficient gait performance (Figure 4). In contrast, correlations were weak in SCZ (*r* < 0.3) and limited in HC to walking speed alone. These patterns imply that motor recovery during abstinence may simultaneously improve upper- and lower-limb coordination in AUD, whereas in SCZ, disrupted cortico-striato-cerebellar connectivity or medication effects may decouple these systems (McGrath et al., 2020).

### Clinical implications and future directions

rHGS and gait metrics are practical, scalable indicators of physical health in psychiatric care (Ogawa et al., 2022). In AUD, they may track abstinence-related recovery (Fein and Greenstein, 2013); in SCZ, they could complement metabolic and cognitive monitoring given the high prevalence of frailty and sarcopenia (Hirjak et al., 2018). However, because this study did not compare wearable-derived measures with clinician ratings or task-based assessments, conclusions about diagnostic or prognostic value remain preliminary. Longitudinal validation against established clinical endpoints will be essential to define their translational utility.

### Strengths and limitations

This study is among the first to apply wearable inertial sensors to evaluate both grip and gait in schizophrenia (SCZ) and alcohol use disorder (AUD). The relatively large clinical sample and phase-specific analysis enhance robustness. By focusing on relative handgrip strength (rHGS) rather than absolute strength, the analysis accounted for body composition and provided greater sensitivity to disease-related decline.

Nonetheless, several limitations must be considered. Cognitive performance was not formally assessed, and subclinical deficits may have influenced the observed motor patterns. Depressive and negative symptoms – known to affect psychomotor speed and motivation – were also unmeasured and could confound performance-based outcomes (Park et al., 2019; Kim et al., 2020; Ganipineni et al., 2023). Accordingly, symptom severity measures (e.g., PANSS or depressive symptom scales) were not available and, therefore, could not be

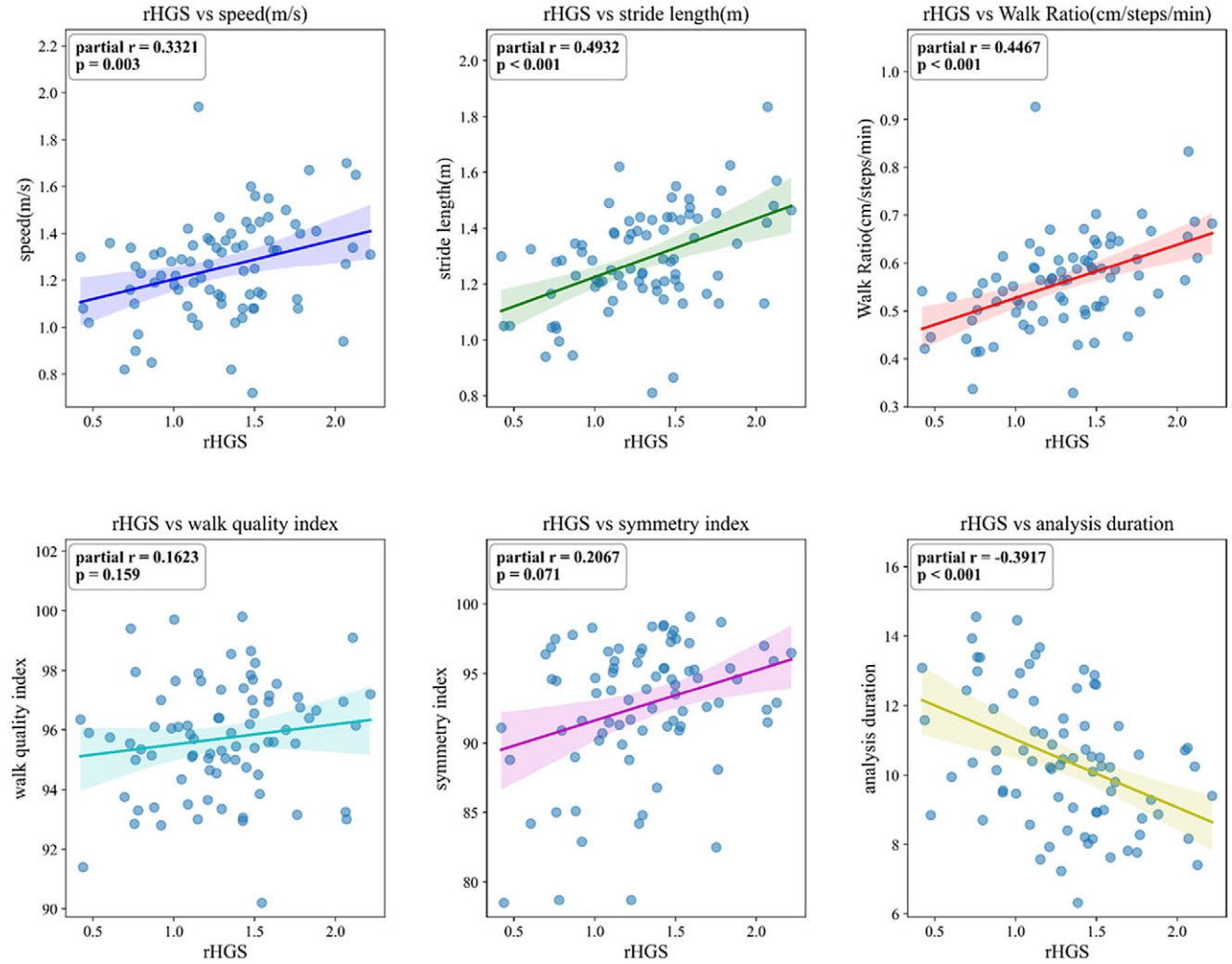

**Figure 4.** Partial correlations between relative handgrip strength (rHGS) and gait parameters in the alcohol use disorder (AUD) group. Comparable results for the healthy control (HC) and schizophrenia (SCZ) groups are provided in Supplementary Figures 1 and 2.

included in the baseline characteristics summarized in Table 1. Demographic imbalance across groups (age, sex and BMI) was statistically controlled but may still contribute to residual confounding. The lack of an upper age limit could further introduce age-related variability.

Although individuals with comorbid SCZ and AUD, nonalcohol substance use or HIV infection were excluded to minimize confounding (Fein et al., 2012; Hunt et al., 2018; Paschali et al., 2024), lifestyle and treatment-related factors – such as smoking, nutritional status, cumulative antipsychotic exposure and physical activity – were not systematically assessed and may have influenced motor outcomes. Because 97% of patients received atypical antipsychotics, which are generally associated with minimal motor side effects (Putzhammer et al., 2004), major medication-induced bias is unlikely; however, residual effects due to polypharmacy or dose variability cannot be completely excluded.

Finally, healthy controls were recruited from nursing students and hospital staff, a convenience sample that may not be representative of the general population. While their occupational activities are classified as light work and unlikely to confer a physical advantage, future studies should include community-based samples with broader demographic diversity to improve generalizability.

## Conclusion

This study demonstrates that wearable sensor–based assessments of handgrip strength and gait capture largely shared motor impairments in schizophrenia and alcohol use disorder compared with healthy controls. In direct comparison between the two psychiatric groups, individuals with schizophrenia showed slightly lower relative handgrip strength than those with alcohol use disorder, while most other gait parameters did not differ significantly between the groups. Notably, schizophrenia was characterized by alterations in gait timing, whereas alcohol use disorder exhibited stronger associations between relative handgrip strength and multiple gait parameters, suggesting differences in motor coordination rather than absolute motor performance.

Although symptom severity was not directly assessed, these findings suggest that motor dysfunction may be partly shared across schizophrenia and alcohol use disorder, while wearable-derived motor metrics may provide useful objective markers for characterizing disorder-specific aspects of functional impairment.

**Open peer review.** To view the open peer review materials for this article, please visit http://doi.org/10.1017/gmh.2026.10190.

**Supplementary material.** The supplementary material for this article can be found at http://doi.org/10.1017/gmh.2026.10190.

**Data availability statement.** The data are not publicly available due to privacy restrictions and Institutional Review Board regulations. Sharing of raw data is prohibited by the ethics committee.

**Acknowledgments.** The authors sincerely thank nurses Kim Hae Ri and Park Soo Kyung for their dedicated assistance in participant care and data acquisition throughout the study.

**Author contribution.** Conceptualization: J.P., B.P. and K.C. Methodology: B.P. and Y.C.K. Data curation: B.P. and J.P. Data visualization: Y.C.K. and B.P. Writing – original draft: J.P., B.P. and Y.C.K. Writing – review and editing: B.P. and H.L. Supervision: B.P. and H.L. All authors approved the final submitted draft.

**Financial support.** This work was supported by INHA UNIVERSITY Research Grant.

**Competing interests.** The authors declare no competing interests.

**Ethical standard.** The research meets all ethical guidelines, including adherence to the legal requirements of the study country.

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
