## [Reviewer Report]

Introduction

1. The manuscript uses “SPR” as the abbreviation for schizophrenia, whereas the standard is usually “SCZ.” In addition, the opening sentence states that “Schizophrenia and alcohol use disorder is associated…,” though the subject is plural. Could the authors clarify and standardize the terminology and correct grammatical inconsistencies?

2. The introduction presents South Korea–specific inpatient statistics and detailed neurodevelopmental vs. neurodegenerative mechanisms. These points seem somewhat tangential to the central rationale. Could the authors consider streamlining these details to keep the introduction more focused on the study’s aims?

3. The discussion of inconsistent findings regarding handgrip strength in AUD is supported by only a few references. Would the authors provide a more comprehensive summary of prior evidence?

4. The introduction identifies gaps (e.g., limited use of rHGS, lack of direct comparisons) but the transition to the study objectives feels abrupt. Could the authors strengthen the linkage between the identified gaps and their specific aims to make the rationale clearer?

Methods

1. The healthy control group included nursing students and hospital staff, who may differ in physical fitness and lifestyle from the general population. Could the authors comment on possible selection bias and how it might affect generalizability?

2. Several extracted metrics (e.g., “walk quality index,” “propulsion index difference”) are not widely used in the clinical gait literature. Could the authors provide clearer operational definitions and references validating these measures to improve interpretability and reproducibility?

3. Numerous gait and grip features were tested (timing, symmetry, quality indices, TUG phases, etc.). It’s not clear whether correction for multiple comparisons was applied.

Discussion

1. Could the authors elaborate on why certain variables lost or gained significance after covariate adjustment, and whether multicollinearity diagnostics were performed?

2. The authors argue that HGS deficits exceed the 1.6 kg threshold considered clinically meaningful. However, their regression coefficients are much larger (–8 kg). Could the authors contextualize whether these large deficits reflect disease-specific pathology versus demographic imbalances, even after covariate adjustment?

3. The authors acknowledge lack of cognitive testing as a limitation. However, other unmeasured confounders (e.g., medication dose variability, substance use history, smoking) may also influence muscle and gait outcomes.

---

## [Reviewer Report]

Summary: This manuscript investigates handgrip strength and gait as assessed objectively by wearable devices as potential biomarkers of schizophrenia and alcohol use disorder. This study found that seven features of grip/gait differentiated the clinical groups from healthy controls, and that grip metrics were correlated with gait metrics. A major strength of the manuscript is its use of a large sample size and multiple clinical groups. Enthusiasm for this manuscript is attenuated by two major factors:

Primary Criticisms:

1. My primary concern is related to the framing of the study:

a. The introduction is quite limited and does not comprehensively review the existing literature. Work from the labs of Sebastian Walther, Vijay Mittal, William Hetrick, and Alexandra Moussa-Tooks have made contributions to the field of sensorimotor function in psychosis, including with wearable/objective measures of grip/gait. Neglecting entire bodies of work in this area results in misleading claims in the introduction that gait/grip are not well studied in psychosis.

b. Accordingly, the discussion does not highlight the importance of this work to the field. One key take-away from the authors is the utility of wearable devices, but this has already been shown (see Walther’s work) and the current manuscript does not test the objective metrics against clinician ratings or other methods, so the authors cannot make this claim.

c. Relatedly, the authors set up the importance of grip/gait as potential biomarkers for psychosis/AUD. More justification for the similarities or differences in these clinical groups would be valuable. There is no comparison between clinical groups.

2. My second concern is the number of statistical tests performed.

a. The rationale for conducting the correlation analyses is weak; what is the utility in knowing that grip and gait are related. Do we only need to use one metric going forward? How might they be related mechanistically?

b. Sex, BMI, and age are used as covariates for the linear regressions, but not for the correlation analysis. I recommend the authors run partial correlations to account for these important covariates.

c. The authors investigate multiple metrics of grip and gait (15 total). In addition, they run univariate and multivariate logistic regressions; this is an immense number of statistical tests. Within process (grip or gait), there is likely to be high correlation and given the correlation between grip/gait, one might argue that a correction for all metrics (Bonferroni=0.05/15=0.003) should be used at least. If so, some of the findings are not significant. Use of effect sizes could be helpful.

Minor Criticisms:

1. Minor grammatical errors throughout; please review.

2. 2 weeks stable on psychotropic medications seems like a short interval; can the authors provide support for this timeframe.

3. It appears there was no consideration or exclusion for depression, which is also linked to significant psychomotor changes and may confound findings.

4. The TUG task should typically be done without an arm rest to limit participants using various levels of assistance; could the authors comment on how participants may have used the arm rests on the chair to assist their standing.

5. When discussing consent, the authors note procedures for illiterate patients. How many illiterate patients were there? Are there group differences in literacy rates? This is important give the links between speech and language (cognitive function) and sensorimotor development, which may confound findings.

6. Fig. 4 should show correlations for all groups (i.e., each group as a line), not just the significant group (AUD).

---

## [Reviewer Report]

The authors included a total of 434 participants (210 healthy controls (HC), 80 with alcohol use disorder (AUD), and 144 with schizophrenia) who underwent iTUG, gait, and hand grip strength (HGS) tests. Different features from the gait and hand grip strengths were compared using linear regression models. The authors show that both people with schizophrenia and alcohol use disorder show lower HGS compared to HC as well as impairments in specific gait and iTUG features while adjusting for sex, age, and BMI. In addition, the authors show that HGS correlates with specific gait features in AUD and to a lesser extent in SCZ. The authors conclude that in particular relative HGS and gait symmetry are the promising biomarkers for motor and cognitive deterioration in psychiatric disorders

It seems like the authors collected a nice dataset with a high number of participants and different objective tests measuring physical capacity. However, I think they missed a great opportunity by not comparing between schizophrenia and AUD using a regression model. In addition, the discussion could be improved by not just summarizing the results but going more into depth what this means and how this could be used for example for future interventions.

Major comments:

- Abstract: The aim should be stated more clearly (e.g., compare between HC and different diagnostic groups). Even though previous studies showed that grip strength can serve as a marker of cognitive function this study does not include any cognitive tests, why the cognitive aspect should be removed here and in the conclusion.

- Introduction: Here, you point towards both distinct and common features between schizophrenia and AUD. You mention the need for “direct, instrumented comparisons across diagnostic groups” and also in your aims in the first point you state you want to compare “across groups”. However, in your regression models you only compare HC-schizophrenia and HC-AUD but not schizophrenia-AUD, which leads to the introduction lacking direction.

- I think this manuscript would greatly benefit from reporting the group comparisons in the regression models between schizophrenia and AUD. This would also be in line with the “comparison between diagnostic groups”. Moreover, there seems to be no study comparing those parameters between the two psychiatric disorders which means doing so would add scientific value.

- Since all age, sex, and BMI are different between groups and are known to influence HGS and gait, I think the univariable model does not make sense and I would suggest to only report the multivariable model. Just for the parameter rHGS the model should only be adjusted for sex and age but not BMI (if not already done so).

- Is there any information available about symptom severity in schizophrenia and AUD (e.g., PANSS for schizophrenia or depressive symptoms)? If yes, I would suggest adding it to table 1 because especially negative symptoms can greatly affect motivation which is required to complete tasks as the ones the authors investigated with the maximal possible performance level. In any case I would discuss this in the discussion and if there is no information available about symptom severity also add this as a limitation.

- The correlations should be controlled for the same variables (age, sex, BMI) as the authors did in the multivariable regression model and a post-hoc test should be added to control for multiple comparisons.

- In the figures showing the correlations there are clear outliers visible in every group. I therefore would encourage the authors to conduct an outlier analysis.

- The discussion/conclusion would greatly benefit from restructuring. For example, I would suggest starting this section with the first paragraph under conclusion which summarizes the results. In general, it seems like the authors are summarizing their results but not discussing what they mean. As mentioned before, here the authors also missed a great opportunity to compare between the two psychiatric disorders and discuss why there are (no) differences in the tests.

Minor comments

- In the abstract, I do not agree that motor function in psychiatric disorders remains underexamined as for example the group of Sebastian Walther has published a lot on this topic. However, if the authors are refereeing to the fact that objective assessment of motor function such as HSG, gait, and iTUG test are not used in psychiatric care, I would suggest rephrasing this accordingly.

- In the conclusion of the abstract, I would suggest being more specific by saying “in schizophrenia and AUD which might extend to other psychiatric populations” instead of just “psychiatric populations”.

- Table 1: I would remain this to “Comparison of demographic and clinical characteristics” as most of them are demographic

- Table 1: I am not sure if there is remission from schizophrenia. It might be better to refer to it as stability period or something similar

- Figure 2: Does the exam duration indicate the mean duration of all participants or were they required to perform this test at a specific speed?

- Figure 3: The p-values between AUD and schizophrenia should be added to make sure it is clear if those groups significantly differ from each other.

- Figure 4: The authors could consider reporting the correlations for all groups together in one figure instead of in the supplementary material by using different colors for each group.

---

## [Reviewer Report]

The topic of the present study is highly relevant, as it addresses the underexplored role of motor function in schizophrenia spectrum and alcohol use disorders. The manuscript is overall well-structured and methodologically sound, with clear clinical implications. Nevertheless, I have a few minor concerns regarding clarity, methodological details, and presentation, which, once addressed, will further strengthen the quality and readability of the work:

1. The abbreviation SCZ is more commonly used than SPR for schizophrenia. Consistency with standard terminology is recommended.

2. In Figure 1, a larger font size is recommended to improve readability.

3. In Figure 3, significant outcomes should be highlighted.

4. The manuscript mentions the potential influence of medication, but it would strengthen the paper if the authors elaborated more explicitly on how medication effects might have biased or altered the results.

5. The clinical groups are considerably older and more male-dominated compared to the healthy control group. Although age, sex, and BMI were statistically adjusted for in the multivariable models, such marked demographic differences may still introduce residual confounding and limit comparability across groups. This issue should be discussed more explicitly in the Limitations section.

6. One of the limitation of the study is that no upper age limit was set for participants, which may have introduced additional age-related effects on motor performance beyond the psychiatric conditions themselves. This should be mentioned in the limitations section.

---

## [Editor Report]

While the manuscript is recognized as timely and methodologically sound, the reviewers identified important issues in framing, analysis, and interpretation (see full reviews below). Key concerns include an unfocused and incomplete introduction, insufficient engagement with prior literature, unclear operationalization of nonstandard gait/handgrip metrics, and lack of correction for multiple comparisons. Several reviewers emphasize the need to add direct comparisons between schizophrenia and alcohol use disorder, adjust associations for covariates, and provide deeper discussion of mechanisms, confounders, and clinical implications. Figures, tables, and terminology also require careful revision for clarity and consistency.

I therefore ask the authors to undertake a thorough revision, carefully addressing all reviewer comments one by one. In particular, please streamline the introduction, incorporate missing literature, justify methodological choices, strengthen the statistical approach and conduct additional analyses as needed, and restructure the discussion to move beyond summary into interpretation and significance. A detailed response letter documenting how each point has been addressed will be required for further consideration.

---

## [Reviewer Report]

Thank you for the revision. The authors have adequately addressed the comments, and the current version is acceptable.

---

## [Reviewer Report]

The authors included a total of 434 participants (210 healthy controls (HC), 80 with alcohol use disorder (AUD), and 144 with schizophrenia) who underwent iTUG, gait, and hand grip strength (HGS) tests. Different features from the gait and hand grip strengths were compared using linear regression models. The authors show that both people with schizophrenia and alcohol use disorder show lower HGS compared to HC as well as impairments in specific gait and iTUG features while adjusting for sex, age, and BMI. In addition, the authors show that HGS correlates with specific gait features in AUD and to a lesser extent in SCZ. The authors conclude that in particular relative HGS and gait symmetry are the promising biomarkers for motor and cognitive deterioration in psychiatric disorders

It seems like the authors collected a nice dataset with a high number of participants and different objective tests measuring physical capacity. However, I think they missed a great opportunity by not comparing between schizophrenia and AUD using a regression model. In addition, the discussion could be improved by not just summarizing the results but going more into depth what this means and how this could be used for example for future interventions.

Major comments:

- Abstract: The aim should be stated more clearly (e.g., compare between HC and different diagnostic groups). Even though previous studies showed that grip strength can serve as a marker of cognitive function this study does not include any cognitive tests, why the cognitive aspect should be removed here and in the conclusion.

- Introduction: Here, you point towards both distinct and common features between schizophrenia and AUD. You mention the need for “direct, instrumented comparisons across diagnostic groups” and also in your aims in the first point you state you want to compare “across groups”. However, in your regression models you only compare HC-schizophrenia and HC-AUD but not schizophrenia-AUD, which leads to the introduction lacking direction.

- I think this manuscript would greatly benefit from reporting the group comparisons in the regression models between schizophrenia and AUD. This would also be in line with the “comparison between diagnostic groups”. Moreover, there seems to be no study comparing those parameters between the two psychiatric disorders which means doing so would add scientific value.

- Since all age, sex, and BMI are different between groups and are known to influence HGS and gait, I think the univariable model does not make sense and I would suggest to only report the multivariable model. Just for the parameter rHGS the model should only be adjusted for sex and age but not BMI (if not already done so).

- Is there any information available about symptom severity in schizophrenia and AUD (e.g., PANSS for schizophrenia or depressive symptoms)? If yes, I would suggest adding it to table 1 because especially negative symptoms can greatly affect motivation which is required to complete tasks as the ones the authors investigated with the maximal possible performance level. In any case I would discuss this in the discussion and if there is no information available about symptom severity also add this as a limitation.

- The correlations should be controlled for the same variables (age, sex, BMI) as the authors did in the multivariable regression model and a post-hoc test should be added to control for multiple comparisons.

- In the figures showing the correlations there are clear outliers visible in every group. I therefore would encourage the authors to conduct an outlier analysis.

- The discussion/conclusion would greatly benefit from restructuring. For example, I would suggest starting this section with the first paragraph under conclusion which summarizes the results. In general, it seems like the authors are summarizing their results but not discussing what they mean. As mentioned before, here the authors also missed a great opportunity to compare between the two psychiatric disorders and discuss why there are (no) differences in the tests.

Minor comments

- In the abstract, I do not agree that motor function in psychiatric disorders remains underexamined as for example the group of Sebastian Walther has published a lot on this topic. However, if the authors are refereeing to the fact that objective assessment of motor function such as HSG, gait, and iTUG test are not used in psychiatric care, I would suggest rephrasing this accordingly.

- In the conclusion of the abstract, I would suggest being more specific by saying “in schizophrenia and AUD which might extend to other psychiatric populations” instead of just “psychiatric populations”.

- Table 1: I would remain this to “Comparison of demographic and clinical characteristics” as most of them are demographic

- Table 1: I am not sure if there is remission from schizophrenia. It might be better to refer to it as stability period or something similar

- Figure 2: Does the exam duration indicate the mean duration of all participants or were they required to perform this test at a specific speed?

- Figure 3: The p-values between AUD and schizophrenia should be added to make sure it is clear if those groups significantly differ from each other.

- Figure 4: The authors could consider reporting the correlations for all groups together in one figure instead of in the supplementary material by using different colors for each group.

---

## [Reviewer Report]

I thank the authors for their thorough response. The concerns raised were overall addressed well and have strengthened the manuscript. Seeing no real differences (with the exception of slight differences in rHSG) between the measured parameters in SCZ and AUD but only observing correlations between rHGS and gait parameters in AUD but not in SCZ is a very interesting finding.

I still have a few minor points, that in my opinion could be improved.

- Results abstract: I would omit the last part “indicating differences in motor coordination patterns across diagnostic groups. Instead, I would add this to the conclusion.

- In the conclusion of the abstract I find the term “distinct yet overlapping patterns of motor impairments in schizophrenia and alcohol use disorder” somewhat confusing. I think it would be clearer to mention the both disorders show reduced HSG and abnormalities in gait features (symmetry index) compared to healthy controls but that motor coordination patterns differ across diagnostic groups and what that means (e.g., in AUD reduced HSG reflects global motor dysfunction, whereas in SCZ this does not seem to be the case/reduced coupling between lower and upper limb function as mentioned by the authors).

- Introduction, page 3, lines 34-36: The sentence “Such discrepancies highlight the need for direct, instrumented comparisons between diagnostic groups using standardized wearable methods” seems out of place, since the previous paragraph talks about inconsistent findings of HSG in AUD and SCZ (or any other diagnostic group) is not yet mentioned.

- Introduction, page 3, lines 51-52: For the second aim I think it would be good to mention what the authors mean by objective FM variables (e.g., gait parameters and/or iTUG test?)

- Information about consent: Although this is an important point, I suggest to only mention it once to avoid repetition (page 4, lines 28-32 under ethical considerations and page 3, lines 48-49)

- In the results section of “clinical characteristics” it would be more informative if the authors would mention the results of a post-hoc test showing which groups actually differed in terms of age, sex, and BMI instead of mentioning numbers already present in table 1.

- Figure 3: Also add (significant) p-values between AUD and SCZ. If space is tight you could also just add significance starts instead of the actual values, which are already in table 2.

- Figure 3: The authors should also add (significant) p-values for differences between SCZ and AUD, specifically for rHSG. Since I understand space is tight, the authors might consider adding significance stars (e.g., * indicates p<0.05, ** p <0.01, *** p<0.001) to significant comparisons only instead of all actual p-values since those are already mentioned in table 2.

- In the first paragraph of the discussion the authors state that both SCZ and AUD show reduces rHSG compared to controls but the two diagnostic groups do not differ from each other. I would suggest changing it to “largely differ” or something similar, because they do show differences, even though barely significant. In addition, I would suggest adding the results from the correlation analysis and briefly mention what this could mean, since I think this is also a very interesting and important observation.

- The conclusion needs to be more in line with the discussion, i.e., in the discussion grip strength is not different between AUD and SCZ, whereas in the conclusion the authors state it is different.

---

## [Editor Report]

The reviewer is overall satisfied with the revision and considers the manuscript substantially improved, but requests a small number of minor clarifications and consistency edits. These mainly concern sharpening the abstract and conclusion, improving clarity and flow in the introduction, avoiding redundancy, adding clearer post-hoc and figure annotations, and aligning the interpretation of rHSG differences and correlation findings across the results, discussion, and conclusion. I therefore ask you to address these minor points and submit a final revised version before the manuscript can be accepted.